Similarities between decapod and insect neuropeptidomes

Veenstra Jan A. jan.veenstra@u-bordeaux.fr
Institut de Neurosciences Cognitives et Intégratives d’Aquitaine (CNRS UMR5287), University of Bordeaux , Pessac , France
Pavasovic Ana
Electronic publication date: 2016 May 26
Publication date: 2016
Volume: 4
Electronic Location ID: e2043
Received 2016 Mar 20; Accepted 2016 Apr 25
Copyright: ©2016 Veenstra
Copyright year: 2016
Copyright holder: Veenstra
License: This is an open access article distributed under the terms of the Creative Commons Attribution License, which permits unrestricted use, distribution, reproduction and adaptation in any medium and for any purpose provided that it is properly attributed. For attribution, the original author(s), title, publication source (PeerJ) and either DOI or URL of the article must be cited.
License URL: https://creativecommons.org/licenses/by/4.0/

Keywords: Neuropeptide, Androgenic insulin-like peptide, Neuroparsin, Agatoxin-like peptide, PDH, Calcitonin, Crustacean female sex hormone, Evolution, Receptor

Funding: CNRS Institutional funding was received from the CNRS. The funders had no role in study design, data collection and analysis, decision to publish, or preparation of the manuscript.

==============================
Background. Neuropeptides are important regulators of physiological processes and behavior. Although they tend to be generally well conserved, recent results using trancriptome sequencing on decapod crustaceans give the impression of significant differences between species, raising the question whether such differences are real or artefacts.

Methods. The BLAST+ program was used to find short reads coding neuropeptides and neurohormons in publicly available short read archives. Such reads were then used to find similar reads in the same archives, and the DNA assembly program Trinity was employed to construct contigs encoding the neuropeptide precursors as completely as possible.

Results. The seven decapod species analyzed in this fashion, the crabs Eriocheir sinensis, Carcinus maenas and Scylla paramamosain, the shrimp Litopenaeus vannamei, the lobster Homarus americanus, the fresh water prawn Macrobrachium rosenbergii and the crayfish Procambarus clarkii had remarkably similar neuropeptidomes. Although some neuropeptide precursors could not be assembled, in many cases individual reads pertaining to the missing precursors show unambiguously that these neuropeptides are present in these species. In other cases, the tissues that express those neuropeptides were not used in the construction of the cDNA libraries. One novel neuropeptide was identified: elongated PDH (pigment dispersing hormone), a variation on PDH that has a two-amino-acid insertion in its core sequence. Hyrg is another peptide that is ubiquitously present in decapods and is likely a novel neuropeptide precursor.

Discussion. Many insect species have lost one or more neuropeptide genes, but apart from elongated PDH and hyrg all other decapod neuropeptides are present in at least some insect species, and allatotropin is the only insect neuropeptide missing from decapods. This strong similarity between insect and decapod neuropeptidomes makes it possible to predict the receptors for decapod neuropeptides that have been deorphanized in insects. This includes the androgenic insulin-like peptide that seems to be homologous to drosophila insulin-like peptide 8.

Introduction

Lobsters, crayfish, prawns, crabs and shrimps are all crustaceans belonging to the order of the decapods. Many of these species are part of the human diet, are sometimes a major source of protein and are often considered a delicacy. While some species are caught in the wild, others, e.g., the freshwater prawn Macrobrachium rosenbergii, are mainly obtained from commercially important cultures. Many of these species are also sufficiently large to allow physiological experiments that are more difficult to perform on smaller arthropods. For these reasons, decapods probably constitute the second best studied group of arthropods after insects. Neuropeptides have also been extensively researched in decapods, and several neuropeptides were initially identified in these crustaceans before they were found in other arthropods such as insects (e.g., Kegel et al., 1989; Stangier et al., 1987; Stangier et al., 1992).

The sizes of their genomes tend to be large (e.g., Yu et al., 2015; Song et al., 2016) and so far no complete decapod genome is available. Initially sequences of crustacean neuropeptides were determined by classical peptide isolation and Edman degradation (e.g., Kegel et al., 1989; Stangier et al., 1987; Stangier et al., 1992; Bungart et al., 1995; Duve et al., 1997), but in the last decade numerous decapod peptides have been sequenced by mass spetrometry (e.g., Dickinson et al., 2008; Dickinson et al., 2009a; Dickinson et al., 2009b; Ma et al., 2008; Ma et al., 2009; Ma et al., 2010; Stemmler et al., 2007a; Stemmler et al., 2007b; Stemmler et al., 2010). In the last two years, identification of the decapod neuropeptidomes has further accelerated using next-generation sequencing methodology. As a consequence we now have fairly long lists of neuropeptides for several decapods. These include Sagmariasus verrauxi (Ventura et al., 2014), M. rosenbergii (Suwansa-Ard et al., 2015), Procambarus clarkii (Veenstra, 2015), Scylla paramamosain (Bao et al., 2015) and Homarus americanus (Christie et al., 2015), while for other decapods significant amounts of data are available to analyze their neuropeptidomes. This is for example the case for Carcinus maenas, Litopenaeus vannamei and Eriocheir sinensis (Li et al., 2012; Ghaffari et al., 2014; Verbruggen et al., 2015; Xu et al., 2015). Some of the ESTs (expressed sequence tags) present in the publicly available databases have been summarized by Christie and his collaborators (Ma et al., 2009; Ma et al., 2010; Christie, 2014; Christie, 2016; Christie & Chi, 2015).

I have previously used the published short read archives for P. clarkii to look for neuropeptide transcripts and could deduce complete or partial sequences for a surprisingly large number of neuropeptide precursors (Veenstra, 2015). When comparing the results obtained in this species, with the lists of neuropeptide transcripts from other decapods, differences appear. While several neuropeptides are consistently found in all species, others are only identified in some. The question is whether these differences are real or represent artefacts. For example, some peptides may not have been searched for in the assembled reads, or there were simply too few reads to allow assembly of a contig, while in other cases the tissue where the particular gene is predominantly expressed was perhaps not included in the analysis. Here I try to answer these questions by reanalyzing published short sequence read archives for a number of decapods.

Materials & Methods

DNA sequences

The following short read archives (SRAs) were downloaded from NCBI using the prefetch command from the SRA Toolkit (http://www.ncbi.nlm.nih.gov/books/NBK158900/): for C. maenas: SRR1564428, SRR1572181, SRR1586326, SRR1589617, SRR1612556, SRR1632279, SRR1632285, SRR1632289, SRR1632290, SRR1632291, SRR1632292 and SRR1632293 (Verbruggen et al., 2015); for P. clarkii: SRR1144630, SRR1144631, SRR1265966, SRR1509456, SRR1509457, SRR1509458 and SRR870673 (Jiang et al., 2014; Tom et al., 2014; Shen et al., 2014; Manfrin et al., 2015); for M. rosenbergii: DRR023219, SRR1559288, SRR345608, SRR572725, DRR023253, SRR1653452, SRR345609, SRR896637, SRR1138560, SRR1653453, SRR345610, SRR896638, SRR1138561, SRR1653454, SRR345611, SRR896645, SRR1138562, SRR567391, SRR896646, SRR1138563, SRR572719, SRR896647, SRR1138564, SRR572720, SRR896649, SRR1138565, SRR572721, SRR896650, SRR1138572, SRR2082768, SRR572722, SRR896651, SRR1138573, SRR2082769, SRR572723, SRR1559287, SRR2082770, SRR572724 (Jung et al., 2011; Ventura et al., 2013; Suwansa-Ard et al., 2015); for S. paramamosain: SRR1310332, SRR1310333, SRR1205999, SRR3086589, SRR834579, SRR1206015, SRR3086590, SRR834580, SRR1310331 and SRR3086592 (Gao et al., 2014; Ma et al., 2014; Bao et al., 2015); for L. vanamei: SRR1037362, SRR1407789, SRR1460505, SRR1952625, SRR2103853, SRR2103860, SRR2895158, SRR1037363, SRR1104812, SRR1407790, SRR1609917, SRR2060962, SRR2103854, SRR2103861, SRR346404, SRR1037364, SRR1105791, SRR1407791, SRR1618514, SRR2060963, SRR2103855, SRR2103862, SRR554363, SRR1037365, SRR1104084, SRR1104085, SRR1460493, SRR1951370, SRR2060964, SRR2103856, SRR2103863, SRR554364, SRR1037366, SRR1184416, SRR1460494, SRR1951371, SRR2060965, SRR2103857, SRR2103864, SRR554365, SRR1039534, SRR1407787, SRR1460495, SRR1951372, SRR2103851, SRR2103858, SRR2103865, SRR556131, SRR1104083, SRR1104080, SRR1104086, SRR1104087, SRR1407788, SRR1460504, SRR1951373, SRR2103852, SRR2103859 and SRR2103866 (Li et al., 2012; Chen et al., 2013; Wei et al., 2014; Gao et al., 2015; Peng et al., 2015); for E. sinensis: ERR336998, SRR1555734, SRR2170964, SRR579530, SRR1199039, SRR1576649, SRR2170970, SRR579531, SRR1199053, SRR1735503, SRR2180019, SRR579532, SRR1199058, SRR1735536, SRR2180020, SRR769751, SRR1199228, SRR1735537, SRR546086, SRR770582, SRR1205971, SRR2073826 and SRR579529 (He et al., 2012; Hui et al., 2014; Li et al., 2014; Sun et al., 2014; Liu et al., 2015; Xu et al., 2015; Cui et al., 2015; Song et al., 2015; Wang et al., 2016); and for H. americanus: SRR2889572 and SRR2891007 (Christie et al., 2015). From Euphausia crystallorophias I analyzed ERR264582 (Toullec et al., 2013) for the presence of a novel putative neuropeptide that was found in the decapod transcriptomes.

The E. sinensis genome was downloaded from http://gigadb.org/dataset/100186, made into a BLAST database and searched for neuropeptide genes as described previously (Veenstra, 2014).

Data analysis

The fasta files were extracted from the SRAs using the fastq command from the SRA Toolkit from NCBI and then made into BLAST databases using BLAST+ (Camacho et al., 2009). Using the P. clarkii predicted neuropeptide precursors, as well as a few other peptides, as queries those databases were then searched using the tblastn command. A few neuropeptide receptors were also analyzed. Identified reads that appeared to belong to the orthologous gene were extracted from the database and then used to identify similar reads using the blastn command. The latter were used as input for the Trinity program (Grabherr et al., 2011) and resulting transcripts were recursively used as input until either the transcript stopped increasing in length or it was judged to be complete based on the location of in-frame stop codons and/or a signal peptide at the N-terminal of the protein predicted from the transcript. Calculations were run on a desktop computer with a AMD FX™-6100 six-core processor and 15.4 Gb of memory under Ubuntu Linux.

This method is very efficient for the extraction of transcripts from single copy genes. However, when there are several paralog genes that have not evolved a lot since their separation, some paralogs may be missed, particularly when their expression levels are low. In those cases, a selection of the particular neuropeptide precursors from which the non-conserved regions (such as the signal peptides) had been removed was used as a query in a tblastn command and all the obtained reads were then fed as input to the Trinity program. It can not be excluded that some less well expressed paralogs of those genes that exist in multiple copies have been missed. This may concern neuroparsin, CHH (crustacean hyperglycemic hormone), PDH (pigment dispersing hormone) and possibly CFSH (crustacean female sex hormone).

Clustal Omega (Sievers et al., 2011) was used for sequence alignments and those were inspected and, when needed, manually corrected using Seaview (Gouy, Guindon & Gascuel, 2010), which was also used to extract the regions for making phylogenetic trees with FastTree (Price, Dehal & Arkin, 2010).

Results

Trinity is a fantastic tool to reconstruct large DNA sequences from very short reads. However, not every sequence corresponds necessarily to a correct cDNA sequence or is biologically interesting. One regularly finds more than one sequence derived from the same gene. In the absence of a genomic sequence, as is the case here, it is not always possible to determine which is the correct one. There are several common causes for the failure to produce a single complete cDNA sequence. First, there may simply be insufficient reads available to produce a complete contig. Secondly, there may be allelic variation that causes the elongation to stop. Thirdly, alternative splicing, as is the case for genes encoding the agatoxin-like peptide, Neuropeptide F 1, CNMamide, calcitonin and CHH, may have the same effect. Fourth, recombining short sequences into a long one becomes very difficult in the case of repetitive sequences. One or more reads containing a sequencing error can aggravate some of the other problems, i.e., lack of sufficient reads, alternative splicing or allelic variants.

Most of the data analyzed here come from natural or almost natural populations that show much larger genetic variation than that found in the typical laboratory animals like mice or rats. Furthermore, many neuropeptide genes code for a number of highly similar neuropeptide paracopies and this makes it no doubt difficult to reconstruct the complete cDNA encoding such precursors and when the various copies are only separated by convertase cleavage sites, the problem may become acute. In one attempt to produce the E. sinensis FMRFamide precursor mRNA Trinity produced a partial transcript that had a perfect internal repeat of 164 nucleotides (Fig. S1), that must have been an artefact; a similar phenomenon is also present in the second predicted orcokinin precursor from S. paramamosain (Bao et al., 2015; Fig. S1). Furthermore, I have previously shown that some neuropeptide genes have alleles that differ in the number of neuropeptide paracopies that they encode (Veenstra, 2010a; Veenstra 2015). It is therefore not surprising that a relatively large number of transcripts for neuropeptide precursors containing multiple paracopies, such as FMRFamide, tachykinin, leucokinin, EFLamide etc, are incomplete even though significant numbers of individual reads are found in the various SRAs. Predictions by Trinity of neuropeptide precursors containing various paracopies may, for the same reasons, contain errors. For example, the allatostatin A precursor from C. maenas does not code for some of the previously identified peptides from this species (Duve et al., 1997), while the Trinity transcripts of several other neuropeptide precursor sequences from the same species that have been obtained by screening of classical cDNA libraries are identical (Klein et al., 1992; Klein et al., 1993; Linck et al., 1993; Chung et al., 2006; Wilcockson & Webster, 2008). Other transcripts that are incomplete are often due to low expression levels.

While this work was in progress a draft genome for E. sinensis sinensis was published (Song et al., 2016). This sequence was prepared using short sequence reads and therefore suffers from the problems associated with this methodology (Richards & Murali, 2015). It is estimated that about 67% of total sequence is present in the current draft. Several of the transcripts identified here are not at all or only partially present in this genome and different exons of the same transcripts are regularly found on different contigs. Its usefulness was, therefore, limited.

The decapod neuropeptide genes that were found are indicated in Fig. 1, where for comparison the presence of neuropeptide genes of Daphnia pulex, a crustacean, and two insects, the termite Zootermopsis nevadensis and the fruit fly Drosophila melanogaster, is also shown. Note that although most neuropeptides have a single name, a few are known by different names, e.g., the natalisins are sometimes called WXXXRamides and some researchers refer to the peptides derived from the neuropeptide like precursor (NPLP) as HIGSLYRamides. Another example is the insect CHH homolog that was initially discovered by its effect on ion transport in the hindgut and is therefore called ion transport peptide (ITP).

Figure 1 Overview of the presence of neuropeptide genes in seven decapods, Daphnia pulex and two insect species.

Dark blue, neuropeptide precursors that have been published previously; light blue, neuropeptide precursors (or significant parts therefore) that can be deduced directly from publicly available TSAs; red, precursors assembled here; yellow, precursors that could not be assembled, but for which individual reads in TSAs demonstrate their existence in the particular species. Asterisks indicate the existence of more than one gene.

Many of the neuropeptide precursor transcripts seem complete, at least as far at the coding region is concerned, while for others very significant parts were found. Since one of the questions raised here is the presence of a particular neuropeptide gene, I have also added neuropeptide genes for which individual reads from an SRA provide evidence for its existence in the particular species, even though Trinity produced no contigs for transcripts from these genes. All the sequences, both DNA and deduced amino acids, are listed in Tables S1–S8.

Distribution

Having all the SRAs it seemed interesting to look at where the various genes might be expressed. Although it is possible to do this for all species involved, some are not very interesting as there is a very limited number of tissues sampled, while in other species the different tissues were sampled on different occasions and analyzed in different fashions, making direct comparisons difficult. However, in the case of C. maenas a single publication reports SRAs for a large variety of tissues (Verbruggen et al., 2015). Therefore, I used this species to look at the expression of the various neuropeptide genes in different tissues. Those neuropeptide receptors for which a contig of a significant size could be obtained and for which a likely ligand could be deduced based on homology to a deorphanized protostomian GPCR (see Veenstra, 2016a) were also included. The actual number of individual reads is often small and quantification of RNAseq reads is tricky due to the PCR amplification protocol used to create these libraries. Furthermore, the actual number of samples remains very small, so care should be taken in the interpretation of these data. Nevertheless, some interesting results are apparent (Table S9). Both the neuroparsins and the CHHs are expressed in virtually every tissue. In the case of the neuroparsin it is the neuroparsin 1 gene that is most abundantly expressed in all tissues, with the other two neuroparsin transcripts present at much lower levels. However, the two identified CHH transcripts are differentially expressed, one hormone is most abundant in the central nervous system and the eye (eyestalk), and the other in the intestine. Other neuropeptides found in the intestine are tachykinin, allatostatin C (Veenstra, 2016b), the B transcript of the calcitonin gene, elevenin, orcokinin, the agatoxin-like peptide and hyrg. The expression of CCHamide 2 and trissin in the hemolymph, presumably in hemocytes, is interesting to note as is the relatively large number of neuropeptides found in the SRA derived from the epidermis.

Paralogs

There are several neuropeptide genes that have one or more paralog genes. These are allatostatin C, CHH, moult inhibiting hormone (MIH), CCHamide, eclosion hormone, neuroparsin, PDH, insulin and CFSH. In some cases these are sufficiently different within the same species and sufficient similar between different species, that they clearly derive from different genes. This is the case for allatostatin C, CCHamide, insulin, neuropeptide F and eclosion hormone.

In the case of PDH, it is a bit more complicated. Variable numbers of PDH precursors were found in the seven decapod species. One group of precursors encoding PDH-like peptides distinguishes itself by an insertion of two-amino-acid residues in the predicted mature PDH. Such a predicted peptide was first found in P. clarkii (Veenstra, 2015), but since it was based on a single read in one species, it seemed premature to give it a distinct name. Now that complete precursor sequences are available and this peptide appears to be ubiquitously present in decapods, I propose to call it elongated PDH, or ePDH, to distinguish it from the more classical forms of these peptides (Fig. 2). The ePDH gene is one of the few genes that is present on a single contig of the draft genome from E. sinensis. It consists of three exons of which the first one is non-coding (Fig. 3). Partial sequence for one of the classical PDH genes show the intron between the two coding exons to be conserved.

Figure 2 Sequence alignment of PDH and ePDH.

Parts of the various PDH precursors including the convertase cleavage sites of the various decapod species. Note that the ubiquitous presence of ePDH that has a two-amino-acid insertion.

Figure 3 Structure of the ePDH gene from Eriocheir sinensis.

The ePDH gene consists of three exons and two introns. DNA sequences coding the signal peptide in yellow, mature ePDH sequence in red and the remainder of the precursor in blue. Numbers indicate sizes of introns and exons in nucleotides. The DNA sequence containing the TATA box and a sequence that is recognizably similar to the Drosophila core promoter motif 1 (in blue, Ohler, 2006) and the start of the mRNA (in red) are also displayed; the red nucleotides at the end are part of the mRNA.

In the case of neuroparsin, PDH, CHHs and its homolog MIH it is not always as clear that they represent different genes with unambiguous orthologs in different species. In some cases the observed differences could reflect allelic variations of a single gene or recent local gene duplications. Although no decapod genomes have been completely sequenced and the E. sinensis CHH genes are mostly very fragmentary, such local gene duplications are well known for CHH in decapods (Gu & Chan, 1998; Gu, Yu & Chan, 2000; Dircksen et al., 2001; Webster, Keller & Dircksen, 2012) as well as Chelicerates (Veenstra, 2016b) and particularly in decapods the number of CHH genes can be quite large (Webster, Keller & Dircksen, 2012).

CHH/MIH

The CHH/MIH neuropeptide family is characterized by CHH, MIH, mandibular organ-inhibiting hormone (MOIH), vitellogenesis-inhibiting hormone (VIH) and gonad-inhibiting hormone (GIH). These hormones have been identifed by different physiological assays, but are in many cases pleiotropic. For example, although identified in different biological assays, VIH and GIH are the same hormone. These peptides can be subdivided in two subfamilies, the CHHs proper and the other peptides. The precursors from the two groups differ in that the CHHs are produced together with a CHH-precursor related peptide, while the prepropeptides from the other homornes consist exclusively of a signal peptide and the sequence of the mature hormone (Webster, Keller & Dircksen, 2012). Four of the CHH/MIH transcripts identified here defy those rules as they do not have the CHH-precursor related peptide, yet on phylogenetic trees they form a separate branch that is closer to the CHH than to the MIH cluster (Fig. 4). Adding more sequences to the tree does not change this (data not shown). In the E. sinensis draft genome many sequences corresponding to these hormones are located on small scaffolds making it impossible to ascertain whether or not these genes are clustered.

Figure 4 Phylogenetic tree showing the evolutionary relationships between the CHH and MIH hormones.

Hormones are those identified from decapod SRAs as well as a few for which the biological activity has been described. Highlighted in yellow are the four sequences that on the tree are more similar to CHH, but lack the precursor-related peptide typically present in CHH.

CFSH

The CFSH is a recent discovery (Zmora & Chung, 2014) and consequently we still know very little of this very interesting hormone. In P. clarkii three related proteins were previously identified (Veenstra, 2015). In five of the other six decapod species two to four such hormones were found, but not in H. americanus, where there are no ovary transcriptomes. The primary sequence of these different putative hormones is not very well conserved, but the cysteine residues are (Fig. 5). The phylogenetic tree of these hormones suggests an initial gene duplication giving rise to two types of CFSH, that I have arbitrarily called CFSH 1 and 2 (Fig. 6). In most species both CFSH 1 and 2 were found, but in L. vannamei only three CFSH 1 paralogs were found and no CFSH 2. In the draft genome of E. sinensis CFSH 1 and 2b genes contain a single coding exon. The CFSH gene 2a transcript is incomplete and it is not clear from the genomic sequence what it is. This hormone was initially isolated from the eyestalk of the crab Callinectes (Zmora & Chung, 2014), while it in the crayfish P. clarkii it seemed to be strongly expressed in the ovary. It seemed therefore of interest to see whether these hormones might be expressed in the ovary in other decapods also. No significant expression was found in the ovaries of M. rosenbergii and L. vannamei (1 to 2 reads maximum for each hormone in an SRA), but 9 reads corresponding to L. vannamei CFSH 1c (as well as 1 each for 1a and 1b) are present in SRR2060962 from the L. vannamei testis. In E. sinensis expression is similar to that in C. maenas (Table S9), high expression levels in the eyestalk and a few reads only in the ovary. For S. paramamosain and H. americanus there are insufficient data to answer this question.

Figure 5 CFSH alignments.

Note that the two types of CFSH are clearly different from one another. Scylla paramamosain is not included in the figure, as there are only a few individual reads from this species. These reads correspond to orthologs of the three crab CFSHs; those that correpond to S. paramamosain 2a predict a protein sequence that is completely identical to the S. olivacea sequence illustrated.

Figure 6 CFSH phylogenetic tree.

Phylogenetic tree of the various CFSH orthologs identified here and elsewhere. The only Scylla sequence is from S. olivacea (GDRN01022056.1). S. paramosain has a very limited number of SRA reads that correspond to three orthologs found in Carcinus and Eriocheir. Note that Macrobrachium, Litopenaeus and Procambarus seem to have independently gone through relatively recent gene duplications.

Neuroparsins and receptors

Three to four neuroparsin transcripts were identified in each of the seven decapod species. Three of the E. sinensis genes were found in the draft genome, two of these (neuroparsins 3 and 4) are on the same scaffold in a tail to tail configuration, where the start and stop codons of the two genes are separated by 11,960 and 9,045 nucleotides respectively (Fig. 7). These are the two E. sinensis genes that are most similar to one another, suggesting that they may reflect the most recent neuroparsin gene duplication in this species. As both these genes seem to have direct orthologs in S. paramamosain and C. maenas, that particular gene duplication possibly occurred in a common ancestor of the three crab species (Fig. 8). The neuroparsin receptor was recently identified as a venus kinase receptor (Vogel, Brown & Strand, 2015); two such receptors are found in all seven decapod species (Table S8). The phylogenetic tree made of the various venus kinase receptors suggests that the other arthropods venus kinase receptors are equally similar to both decapod receptors (Fig. 9).

Figure 7 Configuration of Eriocheir neuroparsin genes 3 and 4.

The relative organization of the two neuroparsin genes relative to one another is indicated. The two genes are located on opposite strands and each gene has four exons and three introns. Numbers indicate the lengths of the exons, introns and the intergenic distance in nucleotides.

Figure 8 Neuroparsin phylogenetic tree.

The different decapod neuroparsin sequences found in the different species were used to make a phylogenetic tree. Note that one neuroparsin gene duplication likely occurred after the crabs separated from the other decapods.

Figure 9 Phylogenetic tree of the tyrosine kinase domains of the decapod insulin and venus kinase receptors.

Venus kinase receptors from the following species were added for increased resolution: Limulus polyphemus, Stegodyphus mimosarum, Locusta migratoria, Ixodes ricinus and Zootermopsis nevadensis. Note that the duplication of the venus kinase receptor gene is not generally present in arthropods and could thus be specific to crustaceans.

Insulin-like peptides and receptors

Three different insulin-related peptides were identified. These are the well known androgenic insulin-related peptide (Fig. 10), an insulin-like peptide (Fig. 11) that seems most similar to the Drosophila insulin-like peptides 1–6 (Nässel & Vanden Broeck, 2016), and a peptide that is orthologous to Drosophila insulin-like peptide 7 and that has been called relaxin (Fig. 12). The latter was previously identified in Sagmariasius and P. clarkii (Chandler et al., 2015; Veenstra, 2015). As can be seen from the figures, the androgenic insulin-like peptide is the least conserved of those three (Figs. 10–12). Insulin-related peptides use two different types of receptors, the typical tyrosine kinase receptor and GPCRs. Insects typical have one gene coding an insulin tyrosine kinase receptor and have one or two GPCRs that are related to the vertebrate relaxin receptors RXFP1 and RXFP2. Given the interest in the androgenic insulin-like peptide both for its intriguing physiology as a peptidergic sex hormone and for its commercial potential (Ventura and Sagi, 2012), I also analyzed the likely insulin receptors.

Figure 10 Sequence alignment of the decapod adrogenic insulin-like peptides.

Note the relatively poor conservation of the primary sequences of these hormones. Conserved residues indicated in black highlighting, and cysteine residues in red.

Figure 11 Sequence alignment of the decapod insulin-like peptides.

Note the much better conservation of the primary sequences of the A and B chains of these hormones. The Carcinus sequence, although incomplete, is clearly part of an insulin precursor. Conserved residues indicated in black highlighting, and cysteine residues in red.

Figure 12 Sequence alignment of the decapod relaxins.

Note the relatively good sequences conservation between the different decapod peptides and Dilp-7. Conserved residues indicated in black highlighting, and cysteine residues in red.

The typical insulin tyrosine kinase receptor, similar to the one recently described from M. rosenbergii (Sharabi et al., 2016), was also found in the other six decapods (Table S8). Two receptors similar to the vertebrate relaxin receptors RXFP1 and RXPF2 were identified. Those two GPCRs are most similar to the Drosophila receptors CG31096 and CG34411, also known as leucine-rich repeat containing GPCR- 3 and 4 (LGR3 and LGR4) respectively. However, the much weaker expression of those receptors made it impossible to deduce their complete cDNA sequences and in some cases no contigs could be obtained. Interestingly the SRA from the E. sinensis accessory gland (SRR2170964) not only shows very large number of reads for the androgenic Insulin-like peptide, but also very significant expression of the insulin tyrosine kinase receptor and a somewhat lower expression of the ortholog of Drosophila LGR3.

Splice variants

There were a number of neuropeptide encoding cDNAs that revealed splice variants. Those that concerned the untranslated regions were ignored, but there are five neuropeptide genes that have alternative transcripts producing different precursors: the CHHs, CNMamide, neuropeptide F 1, calcitonin and the agatoxin-like peptide. In the case of neuropeptide F 1, there is an extra exon sliced into the sequence of the peptide, as described previously from insects and D. pulex (Roller et al., 2008; Nuss et al., 2010; Dircksen et al., 2011). The CNMamide gene in the termite Zootermopsis contains five coding exons, the last two of which are alternatively added to the first three and then produces a different CNMamide-like peptide. In four of the seven decapods, similar alternative splice products were found for the CNMamide precursor. However, while the mature peptide derived from the major splice form is well conserved, the second is much less so (Fig. 13). Two to four splice variants (Fig. S2) were found for the recently discovered μ-agatoxin-like peptide (Sturm et al., 2016). As in some insects (Veenstra, 2014), the calcitonin gene produces two different transcripts, encoding different types of calcitonin, that are similar to the insect calcitonins (Fig. S3). In L. vannamei, M. rosenbergii, H. americanus and P. clarkii the second transcripts are predicted to produce a calcitonin-like peptide that does not have one but two cysteine bridges at is N-terminus (Fig. 14). The calcitonin gene is absent from the E. sinensis draft genome, and hence it is impossible to compare the insect and decapod calcitonin gene structures.

Figure 13 Last parts of CNMamide precursors.

Some arthropods produce alternatively spliced mRNA predicted to produce different CNMamides. Notice that the major splice variants produce a much better conserved neuropeptide than the alternative ones. Residues in red are predicted to be cleaved by convertase and removed by carboxypeptidase during processing; the green glycine residues will be transformed in C-terminal amides and the cysteine residues are orange. Residues conserved between the different species are in blue.

Figure 14 Sequence alignment of the decapod B-calcitonins.

Some of the decapod B-calcitonins are predicted to have two cysteine bridges in the N-terminal part of their sequence, rather than one.

Other peptides

In several cases novel neuropeptides have been detected by mass spectrometry. These are often structural variants of well known neuropeptides such as the RFamides, tachykinins and allatostatins A or B (e.g., Ma et al., 2008; Ma et al., 2009; Ma et al., 2010). However, not all peptide sequences identified this way belong to known neuropeptides. From H. americanus, C. maenas and L. vannamei other peptide sequences have been reported. The ones from C. maenas have previously been suggested to represent fragments of cryptocyanin (Ma et al., 2009), and this was confirmed (Fig. S4). Several of the peptides from H. americanus are shown here to represent fragments of thymosin, actin or histone 2A, however the origins of others remain unclear (Fig. S4). The one peptide reported from L. vannamei, L/IPEPEDPMAEAGHEL/I (Ma et al., 2010), is more interesting, as it could potentially be part of a novel neuropeptide (precursor). This sequence is part of a small protein that has a signal peptide followed by a peptide containing a small piece that is very well conserved (Fig. 15). However, it lacks the classical convertase cleavage sites that one usually finds in neuropeptide precursors and hence its status as a neuropeptide is unclear. Such proteins are also found in the other decapods. Although it was not possible for Trinity to produce a complete contig for S. paramamosain, a similar sequence is present in the databanks for Scylla olivacea. I was unable to find similar proteins in insects, but an orthologous protein was detected in the SRA from Euphausia crystallorophias. The latter sequence shows that the only conserved part is the same as in the decapods (Fig. 15). This peptide was called hyrg (pronounced hirg), for four of the conserved amino acids. Interestingly, the eyestalk seems to be the tissue where expression of hyrg is the highest (Table S9), thus suggesting that it is likely a neurohormone.

Figure 15 Hyrg sequence alignment.

Note that only a small part of the sequence of this puative neuropeptide is conserved in both decapods as well as in Euphausia crystallorophias.

Discussion

Insects and decapods are estimated to have had their last common ancestor about 596 Mya, while similar estimates for the common ancestor of crabs and lobsters on the one hand and that for termites and flies on the other are 322 and 348 Mya respectively (Hedges et al., 2015). The gross morphology of decapods has changed a lot less than that of insects and when one compares the respective neuropeptidomes of those two groups, it is clear that those are similarly much better conserved in decapods than in insects (Fig. 1). Most of the changes in insects are losses of neuropeptides that are particularly pronounced in flies, and perhaps even more so in D. melanogaster.

Whenever in this study a particular gene has not been identified from a decapod species, either one of the following is true: (1) the gene is not expressed at high levels and there are relatively small amounts of RNAseq reads; (2) the gene is expressed predominantly in tissues that have not been sampled in the species in question; or (3) the gene has several paralog genes (PDH, CHH, neuroparsin) and it may not have the same number of paralogs in all species and/or some of those paralogs may be expressed in tissues that were not sampled. A combination of (1) and (2) likely explains the absence of some of the neuropeptide genes in S. paramamosain. From that species the eyestalk was not analyzed, even though this tissue is by far the richest source of neuropeptides. Nevertheless, the existence of several S. paramamosain neuropeptide genes could be inferred from individual RNAseq reads, while the few genes that are completely lacking are only weakly expressed in the other species. The androgenic peptide was found neither in C. maenas nor H. americanus. As in H. americanus only the nervous system was included in the analysis, this is to be expected. In the case of C. maenas, it is plausible that the testis samples did not include the accessory gland, or that the sample was taken at a moment in the life cycle of the animal that expression of this peptide is low or non-existent. With one exception, in all other instances where a transcript seems to be lacking it is either from a gene for which an alternative transcript was found (e.g., in the case of the CNMamide and Neuropeptide F 1 genes), or the number of paralogs may differ between the various species (neuroparsin, CHH, MIH, PDH). The only exception is a typical MIH in H. americanus. In spite of using the MIH sequence of the closely related species H. gammarus (Ollivaux et al., 2006) as a query in the BLAST program, no transcript was found in the H. americanus SRAs. Although MIH has been reported by mass spectrometry from the stomatogastric ganglion of H. americanus (Ma et al., 2008), this peptide is not a typical MIH but has the structure of a CHH (Chang, Prestwich & Bruce, 1990). This suggests that in H. americanus the MIH function is assured by a member of the first rather than the second subfamily of these hormones. It also suggests that in this species the second subfamily was lost.

Neuropeptide evolution

It thus appears that the neuropeptidome of decapods has been remarkably well conserved during evolution. Differences that are found between the insect and decapod neuropeptidomes are the loss or the gain of a neuropeptide. Although there possibly still remain arthropod neuropeptides to be discovered, it appears that the loss of neuropeptides in decapods is limited to a single neuropeptide, i.e., allatotropin. Allatotropin is present in mollusks, annelids as well as chelicerates (Veenstra, 2010a; Veenstra, 2011; Veenstra, 2016a) and hence, it must have been present in the arthropod ancestor. Small peptides are sometimes hard to find using the BLAST program and allatotropin is no exception to this rule (Veenstra, Rodriguez & Weaver, 2012). Nevertheless, as I was neither able to find even a single read correponding to its receptor, including in the very abundant number of transcriptome reads from H. americanus, I conclude that this peptide was most likely lost. In the termite and the fruit fly on the other hand, more neuropeptides are missing, particularly in Drosophila. At first sight insects, as a group, lack EFLamide, the androgenic insulin-like hormone, CFSH and ePDH. However, the recent identification of an EFLamide receptor in Platynereis dumerlii as a TRH GPCR ortholog (Bauknecht & Jékely, 2015) and the presence of such a GPCR in Nilaparvata lugens (Tanaka et al., 2014) suggests that some insects may have such a gene. As described below, it is plausible that the androgenic peptide has an insect ortholog. What seems really different is that many insect species, in particular holometabolous species, have lost several neuropeptides (Derst et al., 2016). Thus Drosophila no longer has genes for elevenin, vasopressin, allatotropin, allatostatin CCC, EFLamide, neuroparsin, calcitonin, ACP, eclosion hormone 2, neuropeptide F 2 and it also lost the possibility to produce alternative transcripts from the CNMamide and neuropeptide F1 genes. The beetle Tribolium castaneum on the other hand still has most of those neuropeptides, but lost allatostatin A, corazonin and leucokinin.

New neuropeptides

Since the last common ancestor of decapods and insects—estimated to have lived 596 Mya (Hedges et al., 2015)—very few neuropeptides seem to have been added to either of the two lineages. Novel neuropeptide genes that have appeared seem all to have originated by duplication from existing ones and are easily recognized as the paralogs of the parent genes. Examples of such genes are the various paralogs of CHH and MIH, PDH and neuroparsin in crustaceans and in insects the typtopyrokinin and SIFamide paralogs as well as the great variety of adipokinetic hormones (all orthologs of crustacean RPCH). The only exception may be hyrg, the precursor for the peptide initially identified from L. vannamei (Ma et al., 2010). This peptide, that is well expressed in the eyestalk and the midgut, has a distribution typical of a neuroendocrine peptide. As I was unable to find it outside of crustaceans, it could be a novel invention of this group. The structure of this putative neuropeptide precursor is somewhat reminiscent of limostatin, a small neuroendocrine protein discovered in Drosophila that intereacts with a GPCR (Alfa et al., 2015) previously identified as the receptor for neuropeptide pyrokinin 1 (Cazzamali et al., 2005). The similarity between limostatin and hyrg resides in the apparent absence of conventional convertase sites in these putative neuropeptide precursors (those postulated to function in the limostatin precursor (Alfa et al., 2015) seem highly unusual (Veenstra, 2000)). In the same context the Drosophila sex peptide comes to mind, as it also acts on a neuropeptide receptor without having neither a well conserved structure nor the typical neuropeptide convertase cleavage sites (Kim et al., 2010). Perhaps one or more of these proteins represent newly evolved ligands for existing neuropeptide receptors that could potentially become novel neuropeptides.

Missing neuropeptides

Many decapod neuropeptides have been identified by mass spectrometry over the years (e.g., Stemmler et al., 2007a; Stemmler et al., 2007b; Stemmler et al., 2010; Christie et al., 2008; Ma et al., 2008; Ma et al., 2009; Ma et al., 2010; Dickinson et al., 2008; Dickinson et al., 2009a; Dickinson et al., 2009b). Most of those were identified in the various SRAs, although not always in exactly the same molecular form. In particular, I was unable to find some of the analogs of SIFamide that have been reported (e.g., Hui et al., 2012). I could neither find [Val1]-SIFamide in any species, however this peptide seems to be present in the stomatogastric nervous system (Christie et al., 2006) and this might explain its absence from the various SRAs. Several of the peptides previously described from these data that did not appear to be neuropeptides could be identified as being part of well known proteins and it also allowed me to identify the hyrg trancript. However, there are three neuropeptides that either have been reported or suggested to be present in decapods that were not found in any of the SRAs from the seven decapod species studied here. These are a pituitary adenylate cyclase activating polypeptide (PACAP) from L. vannamei (Lugo et al., 2013), a GnRH-like peptide from P. clarkii (Guan et al., 2014) and two kisspeptins from M. rosenbergii (Thongbuakaew et al., 2016). None of these peptides could be found in any of the SRAs, neither those from the species from which they were reported, nor from any of the other species. In two cases (PACAP and GnRH), the amino acid sequences of the peptides have been published from the same species used here, so my inability to find these peptides is not due to significant sequence differences between the species used for bioinformatic analysis and those from which the peptides were identified. I was neither able to find evidence for the receptors for such peptides in any of decapods. The GnRH receptor identified from the ovary of the oriental river prawn Macrobrachium nipponense is the corazonin receptor ortholog (Du, Ma & Qiu, 2015), implying that corazonin is its ligand. Given the strong conservation of the decapod neuropeptidome described here, I conclude that is highly unlikely that any of those three peptides is present in decapods.

Functional aspects

Conservation of structure does not necessarily imply conservation of function. The function of crustacean RPCH and its insect ortholog AKH are distinctly different. A neuropeptide sequence does not reveal its function, but the distribution of its receptor may give some clues. FMRFamide is known to effect muscle contraction in decapods (Worden, Kravitz & Goy, 1995), the expression of its putative receptor in muscle, heart and the epidermis (that contains muscle as well) suggests that it has similar effects. The simultaneous expression of elevenin and a putative elevenin receptor in the midgut suggests that is has a digestive function. The hormone GPA2/GPB5 was suggested to be an antidiuretic hormone in Drosophila (Sellami, Agricola & Veenstra, 2011) and was subsequently shown to stimulate sodium reabsorption in the mosquito hindgut (Paluzzi, Vanderveken & O’Donnell, 2014). The very abundant expression of its receptor in the gill suggests that its function C. maenas may well be similar. An interesting difference between insects and decapods is the presence of ecydysis triggering hormone in the decapod nervous system and eye(stalk); in insects this peptide seems to be exclusively present in cells associated with the tracheal system and absent from the central nervous system (Roller et al., 2010). It will be interesting to know whether the function of ecdysis triggering hormone within the decapod nervous system is related to ecdysis behavior.

Intestine

Neuropeptides in the intestine are typically produced by enteroendocrine cells. CHH (Chung, Dircksen & Webster, 1999), SIFamide and tachykinin immunoreactive enteroendocrine cells (Christie et al., 2007) have been previously described from decapods. No SIFamide reads were found in the C. maenas intestine SRA, but allatostatin C, calcitonin-B, elevenin, orcokinin and hyrg were all present in seemingly significant numbers of reads (Table S9). This ensemble of gut neuropeptides differs significantly from what is known from the Drosophila midgut (Veenstra & Ida, 2014), even though tachykinin, allatostatin C and orcokinins are present in both, while the calcitonin B transcript is abundant in phasmid midgut SRAs (Veenstra, 2014).

CHH and MIH

The neuropeptides related to CHH are amongst the best known crustacean hormones (excellent review by Webster, Keller & Dircksen, 2012). As was expected based on the literature, several molecular forms were found. There are reasons to think there may be more of these hormones than reported here. First of all, the few decapod CHH genes that have been identified are typically present in clusters and in Metapaeneus ensis 16 such genes have been found (Gu & Chan, 1998). Secondly, as shown here and elsewhere (e.g., Hsu et al., 2006; Li et al., 2010; Ventura-López et al., 2016) some of these genes are differentially expressed. Thus, if a gene is predominantly expressed in a tissue not included in the analysis, it may not be found. Finally, since these hormones are similar in structure, it is possible that Trinity would have problems producing all contigs.

The biological activities of these hormones vary widely and the hormones with very similar sequences may have quite different physiological effects (e.g., Webster, Keller & Dircksen, 2012; Luo et al., 2015). It is for this reason impossible to interpret the meaning of the four predicted hormones that defy classification as either a CHH-like or MIH-like hormone (Fig. 5).

PDH

There are generally within the same species several precursors coding the shorter, more classical, PDHs, those different precursors code sometimes for the same mature peptide. It seems plausible that some of these differences reflect either allelic variations of a single gene or recent local gene duplications. Most of the species have two or more different predicted mature PDH peptides. It has previously been shown that the two PDHs from the crab Cancer productus have different functions, one is released as a hormone into the hemolymph, while the other is used within the central nervous system (Hsu et al., 2010). As the tissue used for the S. paramamosain transcriptome did not include the eyestalk it is thus not surprising that the hormonal PDH is lacking from the deduced transcriptome in this species. ePDH is not expressed in the eyestalk and one might therefore be tempted to think it is not released into the hemolymph. However, it is present in the L. vannamei transcriptome that was produced from abdominal muscle, hepatopancreas, gills and pleopods (Ghaffari et al., 2014) and thus is likely produced somewhere in the periphery (this transcriptome contains relatively few neuropeptides as it includes neither the central nervous system nor the intestine).

CFSH

CFSH was discovered very recently in the crab Callinectes sapidus (Zmora & Chung, 2014) and consequently we know still very little of this extraordinarily interesting hormone. I previously reported the presence of both CFSH and two homologous proteins in P. clarkii (Veenstra, 2015). Now that there are a few more sequences available, it appears that this gene commonly has several paralogs. Some of these seem to have a relatively recent origin, as the most closely related sequence comes from the same species (Fig. 6). The independent gene duplications of these proteins as well the great sequence variability between and within species may indicate that all these hormones act on the same receptor. Given the relatively large size of these hormones one might expect a leucine rich repeat G-protein coupled receptor or a dimeric protein kinase, perhaps one of the two venus kinase receptors, but this remains speculation. The primary structure of CFSH is not very well conserved and its receptor is unknown. Hence, we don’t know whether an orthologous hormonal regulatory system might be present in other arthropods, like e.g., insects (given the great similarity in their neuropeptidomes this seems a distinct possibility, at least in the more primitive insects). It seems that the expression of this hormone in the ovary of P. clarkii (Veenstra, 2015) is unusual, as it was not found in any of the other decapods for which an ovary SRA is available.

Insulin and neuroparsin

Other intriguing neuropeptides are the neuroparsins and the insulin-related hormones. There are three different insulin-like hormones. There are also three different insulin receptors, the classical tyrosine kinase and two G-protein coupled receptors. What I have called insulin is the hormone most similar to the Drosophila insulin-like peptides 1–6, which function as growth hormones and are also important for reproduction and that signal through the classical tyrosine kinase receptor (Nässel & Vanden Broeck, 2016). The same receptor is also present in decapods as shown here and elsewhere (Veenstra, 2015); it has recently been characterized in two decapods (Aizen et al., 2016; Sharabi et al., 2016). Both insulin and neuroparsins activate tyrosine kinase receptors. However, whereas the actions of insulin in insects are relatively well understood due to very extensive research on these peptides in Drosophila (Nässel & Vanden Broeck, 2016), the function of neuroparsin is less clear, as it is absent from Drosophila melanogaster (Veenstra, 2010b). It is interesting to note that some species have several insulin genes and few if any neuroparsin genes (Drosophila, Acyrthosiphon, Zootermopsis), while decapods and Locusta have several neuroparsin transcripts and only a single insulin gene, suggesting some complementation between these two hormones. Indeed, in some cases, such as vitellogenesis in the mosquito both hormones have synergistic effects (Brown et al., 1998; Dhara et al., 2013), however in the migratory locust they act antagonistically (Badisco et al., 2011). Initially isolated from the migratory locust L. migratoria (Girardie et al., 1989) neuroparsin was shown to have strong anti-juvenile hormone effects, effecting both reproduction and metamorphosis (Girardie et al., 1987). It has been shown that neuroparsin RNAi also inhibits vitellogenesis, and hence reproduction, in the decapod Metapenaeus ensis (Yang et al., 2014). The receptor for this hormone was recently identified in mosquitoes as a venus kinase receptor (Vogel, Brown & Strand, 2015), a type of receptor that was lost in chordates during evolution (Dissous, 2015). Although orthologous venus kinase receptors are present in other arthropod genomes (notably Limulus, Strigamia and Stegodyphus, Table S8) as well as mollusks (Vanderstraete et al., 2013), no neuroparsin orthologs could be found in those species. The evolutionary origin of neuroparsin is therefore unclear and it is not known whether species that seem to lack neuroparsin need a hormone ligand to activate the venus kinase receptor (Dissous, 2015). The presence of two such receptors in decapods is intriguing, but has also been found in Lepidoptera and trematodes (Dissous, 2015).

The other insulin-like peptides

Insects and decapods share many neuropeptides and it is not surprising that the various decapod insulin-related hormones also have insect orthologs. The insulin-like hormone I have called relaxin is an ortholog of Drosophila insulin-like peptide 7. This hormone is not only present in insects, but also in ticks, spider mites, mollusks and even acorn worms (Veenstra, 2010a; Veenstra, Rombauts & Grbić, 2012). As previously pointed out, this hormone is only present in the genomes of those species that also have an ortholog of Drosophila gene CG34411, encoding LGR4 that is homologous to vertebrate relaxin GPCRs (Veenstra, Rombauts & Grbić, 2012; Veenstra, 2014). This suggests that this GPCR functions as a receptor for the arthropod relaxins. It must be noted that this does not exclude the possibility that arthropod relaxins may also signal through the classical insulin tyrosine kinase receptor. In fact, there is evidence from Drosophila that this is so (Linneweber et al., 2014).

Drosophila has an eighth insulin-like hormone that was initially discovered because it is secreted by the imaginal discs (Colombani, Andersen & Léopold, 2012; Garelli et al., 2012). However, data from fly atlas (Chintapalli, Wang & Dow, 2007) show that it is also expressed by the ovary. This hormone was suggested to be acting through the GPCR encoded by Drosophila gene CG31096 encoding LGR3 (Veenstra, 2014) and this has now been confirmed (Vallejo et al., 2015; Garelli et al., 2015). LGR3 is also related to vertebrate GPCRs binding relaxin. As reported previously it has a P. clarkii ortholog (Veenstra, 2016a), and as shown here is generally present in decapods. Combined these data suggest that LGR3 is the receptor for the androgenic insulin-like peptide from the accessory gland. The absence of clear sequence homology between the Drosophila and decapod peptides is not surprising, as the primary sequence of this hormone is poorly conserved in both decapods (Fig. 10) and insects (other insects almost certainly have such a peptide, since they have the receptor, but only within flies is it possible to find orthologs using the BLAST program). Interestingly, both these hormones are produced by gonads or associated accessory glands. At first sight it seems that in crustaceans it is predominantly the male that produces it, while in adult flies it is the female. However, work on the expression of LGR3 in Drosophila shows it to be important for the development of both male and female specific sexual characters (Meissner et al., 2016) and it is perhaps better considered a (sexual ?) maturation hormone for both sexes. This would also make it easier to understand how during evolution it was coopted by the imaginal discs. In decapods the male has two Z chromosomes and is the default sex (Parnes et al., 2003; Staelens et al., 2008; Ventura et al., 2011; Cui et al., 2015). Therefore, one would expect females to have a mechanism (not necessarily hormonal) to escape becoming a male and might thus expect a gynogenic rather than an androgenic hormone (this is one of the reasons why CFSH is so interesting). Even in decapods there is now evidence that the androgenic insulin-like peptide is not specific for males. Thus, in S. paramamosain it is also expressed by the ovary and at higher levels at the end of vitellogenesis (Huang et al., 2014). While the relative levels of expression may seem low as compared to those of actin (Huang et al., 2014), the actual quantities of peptide produced may well rival those made by the accessory gland, considering that the ovary is so much larger (could the accessory gland be the remnants of an embryonic ovary anlage ?). Given the effectiveness of RNAi in crustaceans and the strong phenotypes obtained in the absence of androgenic peptide (Ventura et al., 2009), the hypothesis that LGR3 is important in the transduction of the androgenic peptide signal can be tested. As with relaxin, a GPCR specifically activated by the androgenic insulin-like peptide does not exclude the possibility that it may also act on the classical insulin tyrosine kinase receptor, as shown by recent experiments in the decapod Sagmariasus (Aizen et al., 2016). Possible relations between the decapod insulin-related peptides and their receptors are illustrated in Fig. 16.

Figure 16 Ligand–receptor interactions of insulin-related peptides.

Figure indicates the postulated major interactions of the three decapod insulin-like peptides with three receptors. Secondary interactions are indicated by broken lines. Drosophila gene numbers for orthologous genes are indicated in red. LRR-GPCRs, Leucine-riche repeat GPCRs.

It is of interest to note that the mammalian GPCR most similar to LGR3 is RXFP2, the receptor for insulin-related peptide 3. The latter hormone was initially discovered from the testis and is important not only to insure testicular descent (Adhama, Emmen & Engel, 2000) but also in the female reproductive system (Satchell et al., 2013). Thus the data suggest that not only the structures of the receptor and its ligand are recognizably similar, but so might be their function. This is rather interesting, as most neuropeptides with orthologs in both proto- and deuterostomia have quite different functions in these two groups.

Conclusions

Decapod neuropeptidomes are highly conserved and share many neuropeptides with insects. Although a shared neuropeptide structure does not necessarily translate into a shared function, it should allow for the rapid identification receptors in decapods in those cases where the orthologous insect receptor is known.

Supplemental Information

Supplemental Information 1 Decapod neuropeptide precursors nucleotide coding and deduced protein sequences

Click here for additional data file.

Supplemental Information 2 Tissue distribution of neuropeptides and neuropeptide GPCRs in various tissues of Carcinus maenas.

The number of individual reads found in different SRAs from cDNA prepared from eggs and eleven tissues of Carcinus maenas. Note that the numbers refer to the individual reads corresponding to each gene that are present in each of the twelve SRAs. These numbers are not normalized and as the preparation of cDNA libraries includes a PCR step, such numbers are not a reliable reflection of the expression level of the genes of interest.

Click here for additional data file.

Figure S1 Likely Trinity transcript errors

(A) Trinity generated transcript for Eriocheir FMRFamide precursor. (B) Trinity generated transcript for Scylla orcokinin precursor (Bao et al., 2015). Note that both these contigs have long internal repeats that would be highly unlikely to occur by chance and, hence, suggests that they are artefacts. Nucleotide sequences highlighted in yellow are perfect repeats. (C) Alignment of several crab orcokinin precursors, providing additional arguments to suggest that the second Scylla orcokinin precursor contig is indeed an artefact.

Click here for additional data file.

Figure S2 Alignment of decapod agatoxin-like peptides

Note that both the sequence of the peptide as well as the presence of various transcripts of this gene are well conserved within decapods.

Click here for additional data file.

Figure S3 Phylogenetic tree of arthropod calcitonins

Note that the decapod calcitonins (highlighted in yellow) fit nicely in with the other arthropod calcitonins and are hence easily classified as being either of the A or B type.

Click here for additional data file.

Figure S4 Identification of some peptide sequences found by mass spectrometry in decapods

(A) Deduced amino acid sequence of Carcinus cryptocyanin. Note that the sequence deduced from mass spectrometry data (KIFEPLRDKN) is different from the subsequence in cryptocyanin. However, this may well be an error in sequence interpretation from the mass spectrometry data as KIFEPLRENN and KIFEPLRDKN have very similar theoretical masses (1259.41 and 1259.45 respectively, versus 1259.71 found). This does not explain the KIFEPLVA peptide sequence, but given its similarity to the other peptides, it seems plausible also related to a cryptocyanin. (B) Deduced amino acid sequence of Homarus thymosin containing the subsequence DLPKVDTALK found by mass spectrometry. (C) Deduced amino acid sequence of Homarus histone 2A containing the subsequence AVLLPKKTEKK found by mass spectrometry. The peptide sequence KPKTEKK is perhaps PKTEKK, and if so, it would also be present in histone 2A. (D) Homarus cytoplasmic type 3 actin containing the subsequence LRVAPEEHPVLL found by mass spectrometry. All protein sequences were deduced from SRAs from the respective species using Trinity. Peptides identified within these protein sequences are highlighted.

Click here for additional data file.

I thank Drs. Simon Webster and Haihui Ye for their constructive criticism that helped improve the manuscript. I am also grateful to all those who collected the original data I used, as well as to those who wrote the programs employed for their analysis.

Additional Information and Declarations

Competing Interests

Author Contributions

Data Availability

The author declares there are no competing interests.

Jan A. Veenstra conceived and designed the experiments, performed the experiments, analyzed the data, contributed reagents/materials/analysis tools, wrote the paper, prepared figures and/or tables, reviewed drafts of the paper.

The following information was supplied regarding data availability:

Raw data is available at NCBI in various SRAs, as indicated in the manuscript.

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
