# Peer review of "Similarities between decapod and insect neuropeptidomes"

_PeerJ, doi:10.7717/peerj.2043_

## Round 0.1 · original submission · Minor Revisions

Overall, this manuscript has the potential to make a meaningful contribution to the field and I am in agreement with the reviewers that before it can be accepted for publication, the manuscript would benefit from some minor corrections. Please review all of the comments provided by both reviewers and address them accordingly.

Note the presence of an annotated document from reviewer 1.

·

Basic reporting

This manuscript is (on the whole) well written, clear and unambiguous. previous literature is well cited and relevant. Figures are fine (quite a lot of them though)- but it does represent a very substantial and coherent piece of work. Raw data are available on public databases.

Experimental design

The study involved in silico data mining. Thus,quality of data was outside the author's control. Experimental design appears fine, but one concern I have is that the data from the Carinus transcriptome came from four biological replicates ( numerous tissues). I'm not sure how transcript abundance between tissues was normalized. Also, with just a few biological replicates, influences such as molt stage might have made a significant impact on tissue specific transcript abundance. having said this, these issues were outside the author's control. maybe a caveat could be inserted?

Validity of the findings

No comments

Additional comments

I've made numerous corrections and comments in the pdf version, which I attach. I hope that the author finds these helpful and informative!

·

Basic reporting

At the age of big data, thousands of SRA databases are open access, however, the utilization rate of these data is not enough. Therefore, the re-mining of databases could wake the sleeping data. In this study, author mainly mined neuropeptides from public SRAs in seven decapods and analysed the similarities between decapod and insect neuropeptidomes. This is a meaningful work for bioinformaticians, and somehow could be attractive to the groups of people who are interested in neuropeptides. However, there are some problems should be noticed.
1. In Results, the subheading was named “CCH/MIH” (see line 252), but in Discussion, the subheading was named “CHH and MIH” (see line 501). It will make the readers troubled. If you insist “CCH”, this abbreviation should be explained, otherwise, it will make the other puzzle with “CCHamide”. As I knew, CHH-superfamily can be divided into two subfamilies, type-I and -II according to their structures (reviewed by Webster, 2012).
2. Vitellogenesis-inhibiting hormone (VIH) and gonad-inhibiting hormone (GIH) were named on different occasions, but they are the same hormone in fact. Therefore, “vitellogenesis-inhibiting hormone (VIH) and gonad-inhibiting hormone (GIH)” should be “vitellogenesis-inhibiting hormone (VIH)/gonad-inhibiting hormone (GIH)” (see line 254-255).
3. In insects, ion transport peptide (ITP) belongs to CHH-family peptides. In Figure 1, this feature have been shown, however, the ITP were unmentioned in mainly text.
4. In this study, two Scylla (Scylla paramamosain and Scylla olivacea) were mentioned. But the single word “Scylla” often appears in the main text, figures and tables. This should be specific to species in a situation. On the other hand, the neuropeptidome of Scylla olivacea was entitle “Prediction of Scylla olivacea (Crustacea; Brachyura) peptide hormones using publicly accessible transcriptome shotgun assembly (TSA) sequences” should be noticed.
5. As I knew, HIGSLYRamide and WXXXRamide were identified in decapods. I would to know if these neuropeptides were mined in these seven decapods as well.

Minor comments
1. Line 46, “such as e.g.” should be “such as” or “e.g.”.
2. Line 144-146, the brackets are puzzled.
3. Line 262, “the the CHH” should be “the CHH”.
4. Line 360, line 619, legends of Figure 5 and Figure 6, or anywhere else. “paramosain” should be “paramamosain”.
5. Line 393, “PDF” should be “PDH”, or you should explain why described in “PDF”.
6. Line 516, “(Figs. 5)” should be “(Fig. 4)”
7. Except for “Table 4” and “Table 5”, other three “Table” appear to be “Figure” in sense.
8. Legends of Table 4 and Table 5, what’s the meaning of “eggs and eleven tissues or Carcinus maenas.”? I seems didn’t found the explanation of Table 4 and Table 5 in the main text.

Experimental design

No comments

Validity of the findings

No comments

Additional comments

No comments

---

## Round 0.2 · accepted · Accept

Thank you for addressing the reviewers comments systematically and in great detail. I believe that this manuscript will provide a very meaningful contribution to our understanding of the crustacean biology.